# A Comparison of ^13^C-Methacetin and ^13^C-Octanoate Breath Test for the Evaluation of Nonalcoholic Steatohepatitis

**DOI:** 10.3390/jcm12062158

**Published:** 2023-03-10

**Authors:** Carmen Fierbinteanu-Braticevici, Vlad-Teodor Enciu, Ana-Maria Calin-Necula, Ioana Raluca Papacocea, Alexandru Constantin Moldoveanu

**Affiliations:** 1Department of Internal Medicine II and Gastroenterology—Emergency Hospital Bucharest, “Carol Davila” University of Medicine and Pharmacy, 050474 Bucharest, Romania; 2Gastroenterology Department, Emergency University Hospital, 050098 Bucharest, Romania; 3Physiology Department, Faculty of Medicine, “Carol Davila” University of Medicine and Pharmacy, 050474 Bucharest, Romania

**Keywords:** NAFLD, NASH, Liver fibrosis, methacetin, octanoate, steatosis, breath test

## Abstract

Background: While non-alcoholic fatty liver disease (NAFLD) is a wide-spread liver disease, only some patients progress towards steatohepatitis and cirrhosis. Aim: We comparatively analyzed the methacetin breath test (MBT) for the microsomal function of the liver and the octanoate breath test (OBT) for mitochondrial activity, in detecting patients with steatohepatitis and estimating fibrosis. Methods: 81 patients with histologically proven NAFLD (SAF score) were evaluated. The parameters used for both breath tests were the dose/h and the cumulative dose recovery at multiple timepoints. The statistical association between histological diagnosis and breath test results used Independent Samples *t* Test. The accuracy for diagnosis was evaluated using area under the receiver operator characteristic (AUROC) and the sensitivity and specificity were assessed using the Youden J method. Results: Both MBT and OBT were able to differentiate patients with simple steatosis from NASH and to stratify patients with significant fibrosis and cirrhosis (*p*-values < 0.001 for most analyzed timepoints). The best parameter for NASH diagnosis was OBT dose at 30 min. In the case of significant fibrosis, the most accurate test was MBT cumulative dose at 30 min. Conclusions: Both MBR and OBT tests are potentially useful tools in assessing patients with NAFLD.

## 1. Introduction

While non-alcoholic fatty liver disease (NAFLD) is a wide-spread liver disease, only some patients have progression towards cirrhosis and end stage liver disease (steatohepatitis), while others have a stable non-progressing form (simple steatosis). The risk of progression towards different stages of liver fibrosis is linked to the level of hepatocyte inflammation. Thus, differentiation between simple steatosis (mild/no portal or lobular inflammation) and NASH (ballooning degeneration and lobular inflammation) has been the goal in addressing this risk [1]. The prevalence of NAFLD is estimated at one third of the general population [2].

The current gold standard is the liver biopsy, which can identify the fast progressors and differentiate between NASH and NAFLD [3]. This procedure is, however, an invasive procedure, with significant risks, including bleeding, organ perforation, sepsis, and death. A large retrospective study of 15.181 percutaneous liver biopsies at the Mayo Clinic in 2010 reported 70 cases of bleeding (0.5%) [4]. The largest retrospective European study by Piccinino et al. in 1986 on 68.276 liver biopsies reported cases of bleeding (hemoperitoneum—0.032%, hematoma—0.006%, hemobilia—0.006%, hemothorax—0.022%), infection (0.01%) and perforation (pneumothorax—0.35%, followed by colon, kidney, and gall bladder puncture. Mortality was evaluated at 0.009%, with six total deaths, all in patients with cirrhosis or neoplasia [5]. The procedure is also subject to a number of errors, including sampling and interpreting errors, as well as interobserver variability [6]. The procedure is also rarely accepted by the patient, due to its invasive nature and associated risks.

Numerous tests that identify patients at risk for fast progression were proposed as alternatives to liver biopsy. Two such tests are the methacetin breath test (MBT), which analyzes the microsomal function of the liver, and the octanoate breath test (OBT), which analyzes the mitochondrial activity.

A key component in the fast progression of fibrosis in NASH is the modified hepatic beta-oxidation. Accumulation of fat in the liver results in an increased hepatic beta-oxidation and ketogenesis process [7]. Sodium octanoate is the sodium salt of octanoic acid. It is a medium-chain fatty acid that was initially validated to be used in a non-invasive breath test to measure gastric emptying of solids [8]. Octanoate is absorbed in the digestive tract and transported through the portal system to the liver, where it is metabolized by beta-oxidation to acetyl-CoA and CO_2_ [9].

Methacetin, a derivative of phenacetin, goes through the hepatic oxidase system in the liver mitochondria, where it is metabolized to acetaminophen and carbon dioxide [10]. Previous studies have shown that methacetin can reliably differentiate between healthy controls and patients with cirrhosis [11,12,13,14,15]. Some studies have also previously tried to assess the changes in mitochondrial activity with changes in NASH [16].

Both substances are metabolized by the liver—tied to mitochondrial activity (in the case of methacetin) or tied to beta-oxidation (in the case of sodium octanoate). Both metabolisms have CO_2_ as an end result, and by marking the CO_2_ with an isotope (^13^CO_2_), we can measure the exact amount resulting from the metabolism we are trying to evaluate and estimate the mitochondrial activity and beta-oxidation of the liver.

### Aim

The aim of our study was to comparatively analyze the two liver assessment methods and determine which is more accurate in detecting patients with steatohepatitis and estimating fibrosis.

## 2. Materials and Methods

### 2.1. Study Design

We performed a prospective study on patients with non-alcoholic steatohepatitis, between October 2014 and September 2018. A study protocol was applied for each patient, which included a standard clinical exam, blood tests (including hematology, liver function tests, coagulation tests), abdominal ultrasound, ^13^C-Methacetin Breath test, ^13^C-Octanaote Breath test, and liver biopsy. All tests were performed within a maximum of 1 week within each other (usually within 2–3 days), except for liver biopsy, where a recent biopsy within a maximum of 1 year was accepted.

### 2.2. Clinical Exam

The clinical exam consisted of a comprehensive analysis, including medical history, current symptoms, current treatments and allergies, alcohol consumption and abuse assessment (CAGE questionnaire). The physical exam included measurements of height, weight, waist circumference, blood pressure, heart rate, cardiac rhythm regularity, respiratory rate. Measurements were performed with the patient at rest for at least 10 min. Body mass index (BMI) was calculated (kg/m^2^) as weight (kg) divided by height (m^2^), and waist circumference (WC) was measured at the midpoint between the lower border of the rib cage and the iliac crest.

### 2.3. Blood Tests

A standard hemogram was assessed in all patients, including evaluation of thrombocytes, leukocytes, hemoglobin, red blood cell characteristics (mean corpuscular volume and mean hemoglobin concentration).

Biochemistry tests included both transaminases (alanine aminotransferase—ALT, aspartate aminotransferase—AST), serum albumin and serum total proteins, cholestasis enzymes (γ-glutamyl transpeptidase—GGT, total bilirubin, direct bilirubin, alkaline phosphatase), lipid profile (including triglycerides, total cholesterol, high density lipoprotein cholesterol, low density lipoprotein cholesterol), uric acid, renal function tests (urea, creatinine).

Coagulation tests included prothrombin time (PT), international normalized ration (INR), activated partial thromboplastin clotting time (APTT).

Other tests performed included screening for chronic viral hepatitis (HBs antigen, HCV antibodies), liver carcinoma (alfa-fetoprotein), and other types of chronic hepatitis (anti-mitochondrial antibodies, serum ceruloplasmin, antinuclear antibody, alfa1-antitripsin).

Devices used to analyze the samples included CELL-DYN 370 (Abbot Diagnostics, Chicago, IL, USA), ARCHITECT c 8000 (Abbot Diagnostics, Chicago, IL, USA), ACL TOP 500 (Instrumentation Laboratory, Bedford, MA, USA), Access 2 Immunoassay System (Beckman Coulter, Brea, CA, USA), Dimension RXL analyzer (Siemens, Erlangen, Germany).

### 2.4. Abdominal Ultrasound

Hepatic steatosis was assessed using the four-point scale of hyperechogenicity: 0 = absent, 1 = light, 2 = moderate, 3 = severe, according to the difference in density between the liver and the right kidney. Spleen volume was estimated by measuring the spleen longitudinal diameter (SLD) corresponding to the maximum length between the two poles of the spleen. The device used was Acuson S2000 (Siemens AG).

### 2.5. Liver Biopsy

Liver biopsy was performed in all patients, within a maximum of 1 year of the other tests. The procedure was performed by senior physicians, using the Menghini technique with a 1.4 mm diameter needle (Hepafix; Braun, Germany).

Histopathological analysis was performed by an expert (20-year experience) blinded to the patients’ clinical results. Liver biopsy samples accepted for histological assessment had a minimum length of 20 mm and 8 portal tracts.

The SAF classification and FLIP Diagnosis Algorithm were used to establish positive diagnosis [17].

### 2.6. Breath Testing Protocols

All breath tests were performed after an overnight fast. During the test, any food consumption and physical activity were prohibited. To avoid cross-contamination, the two breath tests were performed on different days, but no longer than 1 week apart.

For each test, a substrate was administered orally (for the MBT test—100 mg methacetin labeled with stable, non-radioactive isotope ^13^C solved in 200 mL water, for the OBT test—100 mg octanoate labeled with stable, non-radioactive isotope ^13^C solved in 200 mL water). A timer was started at the moment of substrate administration, and each patient was asked to exhale at regular intervals into specific collection bags. Each bag was labeled with the patient study ID number, substrate, and time from administration.

In both cases, the first bag was collected before substrate administration (labeled as 0 min), to serve as control. Subsequently, samples were collected at regular intervals: 10, 20, 30, 40, 50, and 60 min from substrate administration for MBT and 15, 30, 45, 60, 120 min from substrate administration for OBT.

The analysis was performed using a ^13^C/^12^C Infrared Spectrometer (IRIS Doc, Wagner), within a maximum of 60 min from the last sample being collected. Parameters measured by the device included ^13^CO_2_ recovered as a function of time (PDR [%/h]) and cumulative exhaled ^13^CO_2_ (cPDR [%]). The analyzer performed all the measurements for a single patient and a single substrate simultaneously and calculated the PDR and cPDR automatically for each point in time.

### 2.7. Inclusion Criteria

The main inclusion criteria consisted of patients of at least 18 years of age, willing to participate in the study (signed informed consent), with newly established or previously established (within the last 12 months) diagnosis of non-alcoholic fatty liver disease (by histological analysis of liver biopsy sample).

### 2.8. Exclusion Criteria

Patients unwilling to participate in the study at any point in time were excluded from the study.

An unclear or uncertain liver biopsy result was also considered an exclusion criterion. Other causes of chronic liver disease with necroinflammatory activity also led to the exclusion of the patient from the study: chronic viral hepatitis, alcohol induced liver disease, high alcohol consumption or alcohol abuse (CAGE questionnaire, alcohol consumption estimation of more than 20 mg/day for women and 30 mg/day for men, indirect markers—increased mean corpuscular volume, isolated increase in gamma-glutamyl transpeptidase), hemochromatosis, autoimmune hepatitis, primary biliary cirrhosis, Wilson’s disease, hepatocarcinoma.

The patient’s chronic drug administration was given special interest, and drugs that could cause drug-induced liver disease (highly liver toxic drugs), or that interfered with either the methacetin or octanoate metabolisms, or that could independently cause NAFLD, were carefully analyzed. Patients who required chronic administration of these drugs were excluded from the study. Such drugs included: amiodarone, corticosteroids, methotrexate, stavudine, tetracycline, valproic acid, zidovudine.

Comorbidities that excluded patients from the study included severe COPD (GOLD C or above) and severe asthma as these could potentially interfere with the elimination of the ^13^CO_2_ as well as any type of malabsorption syndrome as these could potentially interfere with the bioavailability of the substrate. Other comorbidities that were considered exclusion criteria included: uncontrolled diabetes (HbA1C > 7%), severe congestive class failure (NYHA class 3 or above)—Appendix A.

Other exclusion criteria included pregnancy, participating in other clinical trials, hypersensitivity to any of the substrates, significant weight change during the study protocol (defined as >10%), and recent acute disease that required medical or surgical treatment (past 3 months).

### 2.9. Study Groups

Patients were divided into three groups: group 1 included patients with simple steatosis (liver biopsy result negative for steatohepatitis), group 2 included patients with steatohepatitis but low-grade fibrosis (liver biopsy result positive for steatohepatitis, with F score 0 or 1), and group 3 included steatohepatitis with significant fibrosis (liver biopsy result positive for steatohepatitis, with F score of at least 2). For diagnosis accuracy analysis, group 1 was compared with both groups 2 and 3 together, while for significant fibrosis, diagnosis group 3 was compared with groups 1 and 2 together.

### 2.10. Statistical Analysis

The results were entered into a table using Microsoft Excel (Microsoft corporation) and analyzed using Excel (Microsoft Corporation), Python with the Pandas, Statsmodels and Scipy libraries, and SPSS version 23 (IBM corporation). Normality of the variables was assessed using histograms and the Shapiro–Wilk test. In univariate analysis, the independent samples *t* test was used to assess statistical difference. Crosstabulation analysis for sex was performed using the chi-square test. Bivariate analysis was performed to assess the strength of the association between breath test results and SAF score inflammation using Spearman’s rho test. The receiver operating characteristic (ROC curve) was constructed for the best parameters and an optimal cutoff was chosen using the Youden j statistic method. Sensitivity, specificity, positive predictive, and negative predictive values were calculated. The overall accuracy was calculated using the area under the ROC curve (AUROC).

## 3. Results

### 3.1. Characteristics of Patients

The study initially included 159 patients, of which 78 were excluded due to exclusion criteria. The remaining 81 patients were divided as follows: Group 1 (simple steatosis)—31 patients, Group 2 (NASH with low grade fibrosis)—11 patients, Group 3 (NASH with significant fibrosis)—39 patients.

### 3.2. Univariate Analysis

Several factors were associated with non-alcoholic steatohepatitis, including aspartate and alanine aminotransferases (*p* < 0.001 and *p* = 0.006, respectively), gamma-glutamyl transpeptidase (*p* < 0.001), triglycerides (*p* < 0.001), serum albumin (*p* < 0.001). Regarding the breath tests, for MBT significant values were the PDR (%/h) at 10 min, 20 min, from substrate administration, as well as the cPDR in all periods from substrate administration (*p* < 0.001). In the case of OBT, significant values were the PDR (%/h) at 15 min, 30 min, 45 min, and 60 min from substrate administration, as well as the cPDR at most timepoints from substrate administration (*p* < 0.05). The best parameter for NASH diagnosis was OBT PDR at 30 min—average values 22.08 (NASH) vs. 17.45 (steatosis), *p* < 0.001. Regarding significant fibrosis, most parameters followed a similar pattern, with MBT PDR having correlations in the initial phases (10–20 min, *p* < 0.001), MBT PDR and OBT being correlated across most timepoints (*p* < 0.05). The most accurate test was MBT cumulative dose at 30 min—average values: 7.55 (significant fibrosis) vs. 12.64 (non-significant fibrosis), *p*-value < 0.001. Extended data, grouped by each fibrosis category (rather than advanced vs. non-significant fibrosis) are listed in Table 1 and Table 2. A crosstab between fibrosis and activity is provided in Table 3.

For MBT, the bivariate analysis identifies a strong negative correlation in PDR in the early phases (r = −0.51 at 10 min), as well as strong negative correlation in cPDR in the entire range (r between −0.50 and −0.59), with the maximum r = −0.59 at 40 min. For OBT, the bivariate analysis reveals a strong positive correlation in the initial phases of PDR (r = 0.61 at 20 min), and a good cPDR in the later phases (r = 0.49 at 60 min).

### 3.3. Diagnostic Performance of the ^13^C Breath Test

Regarding NASH diagnosis, the area under the ROC was best for OBT PDR at 30 min, with an overall area under the curve of 0.855 (95%CI 0.766–0.943). The difference in PDR for steatosis vs. NASH for octanoate administration at multiple timepoints is shown in Figure 1. Using a cutoff value of 19, the test yielded a sensitivity of 80%, specificity of 83%, positive predictive value of 85%, and negative predictive value of 77%. By contrast, MBT in NASH diagnosis achieved a more modest area under the curve of 0.825 (95%CI 0.733–0.927). Using a cutoff value of 10, the test yielded a sensitivity of 75%, specificity of 94%, positive predictive value of 94%, and negative predictive value of 76%. 

Regarding significant fibrosis, the area under the ROC was best for MBT cPDR at 30 min, with an overall area under the curve of 0.891 (95%CI 0.798–0.984). The difference in cPDR for methacetin administration, for steatosis vs. NASH with significant fibrosis at multiple timepoints is shown in Figure 2. Using a cutoff value of 11, the test yielded a sensitivity of 92%, specificity of 90%, positive predictive value of 90%, and negative predictive value of 93%. Figure 1 By contrast, OBT (cPDR at 30 min) in significant fibrosis achieved a more modest area under the curve of 0.783 (95%CI 0.683–0.883). Using a cutoff value of 8, the test yielded a sensitivity of 66%, specificity of 86%, positive predictive value of 81%, and negative predictive value of 73%. Area under ROC is provided in Table 4 and a DeLong test was performed for NASH diagnosis and significant fibrosis, between the two tests, with results provided in Table 5.

## 4. Discussion

NAFLD is a continuum of fatty liver injuries characterized by interconnected histological, functional, and metabolic derangements of the mitochondria consisting of ultrastructural lesions, crystalline mitochondrial inclusions, increased mitochondrial volume, reduction of the mitochondrial cristae [16].

These evolutive and dynamic changes are translated into disfunction of the respiratory chain, followed by reduced energy production and increased oxidative stress [18].

The aim of the present study is to provide a comparison in terms of significance and accuracy of two noninvasive metabolic tests, used to stratify the patients with fatty liver disease. Moreover, these tests were considered useful tools for the stratification of the prognosis in NAFL patients, being surpassed only by liver biopsy. A supplementary advantage of these tests are the real-time evaluation results, allowing a fast clinical intervention when necessary. Both ^13^C MBT and OCT belong to the category of dynamic tests able to assess the function of the liver. As long as the ^13^C MBT evaluates the microsomal function, it is considered a functional mass of the liver and proved to be the highest accuracy on long-term evaluation of the liver disease prognosis [12].

In the present study, we demonstrated that both OBT and MBT were able to differentiate patients with simple steatosis from those with NASH.

We identified the OBT PDR at 30 min as the best parameter for NASH diagnosis—average values 22.08 (NASH) vs. 17.45 (steatosis) *p* < 0.001, even though there were significant values of the PDR (%/h) at 15 min, 30 min, 45 min, and 60 min from substrate administration, and of the cPDR at most timepoints after OCT ingestion (*p* < 0.05).

Our results confirm previous data where we showed that ^13^C-OBT can identify the patients with NASH through their ability to better/faster metabolize octanoic acid [19].

This increase of the octanoate metabolization, followed by the ^13^CO_2_ elimination in the exhaled air which is expressed as cumulative percentage of recovery—cPDR, was also confirmed in a previous study of our group [20].

Similar results regarding the higher terminal oxidation of the ^13^C-octanoate were reported in patients with the NASH versus control group [18,21].

To evaluate the efficacy of OBT in differentiating between patients with NASH and patients with NAFL, we analyzed the area under the ROC curve, which revealed that the PDR at 30 min has the best diagnostic power for NASH—0.855 (95%CI 0.766–0.943 for a cutoff value of 19, with a sensitivity of 80%, a specificity of 83%, a positive predictive value of 85%, and negative predictive value of 77%.

^13^C-methacetin suffers the process of O-demethylation in the liver’s microsome, resulting in ^13^CO_2_. ^13^C-MBT was reliable for the discrimination between patients with and without significant fibrosis and cirrhosis [22].

In our study, when we talked about the significant fibrosis, MBT PDR and OBT were correlated across most timepoints (*p* < 0.05); however, the most accurate test was MBT cumulative dose at 30 min, with an average of 7.55 in significant fibrosis vs. 12.64 in non-significant fibrosis, *p* < 0.001. The area under the ROC was best for MBT cPDR at 30 min, with an overall area under the curve of 0.891 (95%CI 0.798–0.984).

^13^C MBT was better than ^13^C OCT in stratifying patients with significant fibrosis and cirrhosis (*p*-values < 0.001 for most timepoints analyzed).

Recent data correlate the excessive fat storage to increased liver stiffness and subclinical microsomal liver dysfunction, which can be detected by ^13^C MBT. The importance of this finding is the potential identification of increased portal resistance in the absence of fibrosis, due to the higher stiffness of the lipid droplets in comparison to hepatocytes’ cytoplasm [23], however, this will need to be investigated in future studies. Other potential new directions for research could include the application of artificial intelligence (AI) to these parameters, as some AI algorithms have already been applied to the histopathological exam [24].

Our study has a few limitations. The exclusion criteria were chosen to, as accurately as possible, assess the underlying metabolic process, with as little interference as possible from other pathologies or complications. Unfortunately, in practice, this means the study has not assessed patients that have common comorbidities with steatohepatitis, including diabetes and other elements of metabolic syndrome. Another significant limitation of the study was the relatively small number of patients included in the study. While for most patients, the biopsy was performed within a few weeks of the other tests, for some, the difference between the biopsy and the other tests was as long as one year.

## 5. Conclusions

Both MBR and OBT tests provide reliable data, with OBT having a slightly higher overall accuracy for activity, while MBT having higher accuracy for fibrosis. Both tests are also non-invasive, potentially useful tools in assessing patients with NAFLD.

These findings create new challenges for the role and value of the metabolic breath tests, with key implications in clinical hepatology. The sequence and algorithm of combined tests use may bring new data for the NAFLD pathogeny and evolution.

## 6. Patents

This section is not mandatory but may be added if there are patents resulting from the work reported in this manuscript.

## Figures and Tables

**Figure 1 jcm-12-02158-f001:**
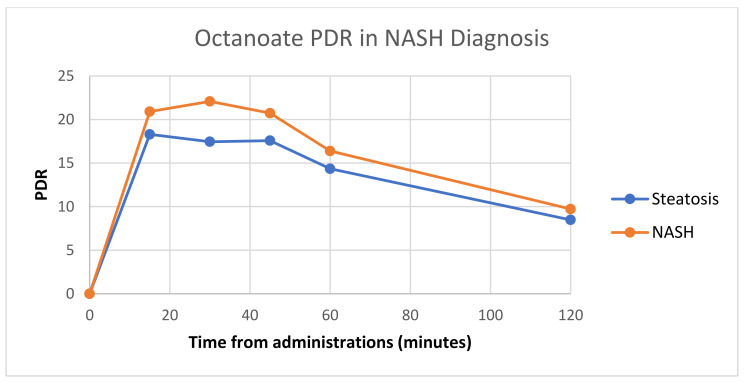
Comparison of PDR in steatosis vs. NASH for octanoate administration.

**Figure 2 jcm-12-02158-f002:**
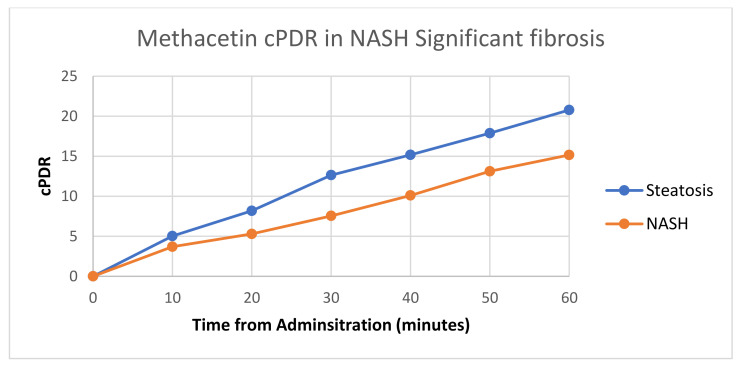
Comparison of cPDR in steatosis vs. NASH for methacetin administration.

**Table 1 jcm-12-02158-t001:** Extended patient characteristics, by fibrosis grade.

Variable	F0	F1	F2	F3	F4	Method	*p*-Value
Number	32 (40%)	10 (12%)	8 (10%)	12 (15%)	19 (23%)		
Age	44.62 (SD = 13.45)	57.1 (SD = 12.52)	50.25 (SD = 9.04)	51.83 (SD = 7.31)	54.21 (SD = 11.46)	ANOVA	0.016
BMI	28.81 (SD = 6.2)	31.5 (SD = 6.55)	30.88 (SD = 6.49)	30.58 (SD = 4.42)	31.16 (SD = 5.53)	ANOVA	0.575
Glucose	110.0 [66–173]	106.0 [71–124]	118.0 [95–153]	112.0 [92–143]	138.0 [69–205]	KW	0.170
HOMA	2.9 [1.7–3.89]	2.84 [2.4–3.09]	2.86 [2.12–3.5]	3.24 [2.76–4.34]	3.28 [2.27–5.19]	KW	0.027
Triglycerides	167.0 [99–315]	185.0 [105–293]	186.0 [152–347]	200.0 [123–327]	201.0 [74–302]	KW	0.073
Cholesterol	226.0 (SD = 95.0)	191.0 (SD = 46.0)	200.0 (SD = 35.0)	227.0 (SD = 74.0)	230.0 (SD = 51.0)	ANOVA	0.590
HDL	35.25 (SD = 7.45)	39.4 (SD = 2.76)	33.5 (SD = 13.72)	35.08 (SD = 7.32)	33.53 (SD = 7.56)	ANOVA	0.404
ALT	64.0 [21–133]	61.0 [40–204]	100.0 [46–158]	96.0 [58–126]	99.0 [59–157]	KW	0.002
AST	74.0 [48–127]	74.0 [51–114]	84.0 [53–116]	80.0 [66–138]	86.0 [51–134]	KW	0.349
ALB	4.0 [3–5]	3.0 [2–5]	3.0 [3–4]	3.0 [2–4]	3.0 [2–4]	KW	0.002
INR	1.0 [0.8–1.3]	1.1 [0.9–1.2]	1.0 [0.9–1.2]	0.95 [0.9–1.3]	1.0 [0.9–1.1]	KW	0.388
γGT	78.0 [38–185]	96.0 [41–127]	103.0 [84–192]	119.0 [78–233]	114.0 [69–279]	KW	<0.001
FERITIN	147.0 [114–228]	145.0 [111–185]	134.0 [106–264]	160.0 [127–390]	177.0 [129–284]	KW	0.065
CRP	1.62 [0.6–6.11]	3.54 [1.58–7.03]	4.54 [1.99–8.46]	5.16 [3.6–7.8]	5.52 [3.25–8.33]	KW	<0.001
URIC ACID	5.07 [3.06–8.36]	4.92 [4.24–9.85]	6.18 [2.84–8.65]	7.27 [5.53–9.35]	6.36 [4.58–8.6]	KW	<0.001
Spleen Diameter	105.0 (SD = 12.0)	108.0 (SD = 10.0)	111.0 (SD = 10.0)	121.0 (SD = 11.0)	124.0 (SD = 15.0)	ANOVA	<0.001
Abdominal circumference	91.0 [65–104]	91.0 [82–107]	95.0 [74–131]	94.0 [76–131]	92.0 [64–110]	KW	0.697
Sex	21 F (26%)/11 M (11%)	7 F (9%)/3 M (4%)	6 F (7%)/2 M (2%)	3 F (4%)/9 M (11%)	8 F (10%)/11 M (14%)	Chi	0.054
MDose10	32.32 [27.68–43.06]	33.29 [10.67–41.45]	24.7 [16.83–43.77]	10.83 [5.36–23.67]	9.92 [6.48–43.44]	KW	<0.001
MDose20	29.56 [23.73–36.31]	30.41 [8.73–36.45]	30.74 [10.1–52.75]	10.87 [9.53–31.68]	11.1 [8.35–32.0]	KW	<0.001
MDose30	23.64 [20.12–27.61]	26.03 [14.16–30.57]	29.3 [13.63–32.15]	14.3 [12.66–35.1]	14.72 [11.81–31.77]	KW	0.001
MDose40	19.35 [16.08–24.35]	20.74 [17.08–27.91]	26.12 [17.0–32.33]	18.97 [14.58–30.87]	16.99 [13.66–27.63]	KW	0.014
MDose50	16.02 [13.71–17.99]	16.74 [13.9–23.25]	20.96 [13.4–24.79]	13.9 [11.47–25.94]	13.43 [11.34–24.27]	KW	0.002
MDose60	13.62 [11.85–15.23]	13.66 [10.23–20.32]	15.16 [10.69–20.15]	11.25 [8.85–17.94]	11.61 [8.58–16.41]	KW	0.001
cMDose10	5.21 [4.09–6.2]	5.02 [2.33–6.11]	3.47 [2.74–10.18]	3.08 [2.52–3.52]	2.92 [2.51–8.06]	KW	<0.001
cMDose20	8.52 [6.34–10.82]	7.78 [3.36–10.12]	5.42 [3.58–13.7]	4.01 [3.47–5.31]	4.9 [3.12–12.17]	KW	<0.001
cMDose30	13.33 [10.3–15.15]	12.54 [3.96–14.6]	8.36 [5.75–21.61]	5.76 [4.12–9.93]	6.24 [3.34–16.64]	KW	<0.001
cMDose40	15.46 [12.79–17.5]	15.58 [4.82–17.96]	10.36 [9.26–25.61]	8.24 [6.25–11.05]	8.49 [5.33–21.61]	KW	<0.001
cMDose50	18.12 [14.87–21.17]	18.28 [7.33–20.49]	13.15 [12.09–26.96]	11.28 [8.89–14.83]	11.25 [8.76–25.17]	KW	<0.001
cMDose60	21.26 [17.7–24.83]	20.77 [7.94–24.58]	14.18 [13.25–30.66]	12.09 [9.72–18.92]	13.8 [9.27–28.44]	KW	<0.001
ODose15	20.26 [8.39–24.68]	16.75 [13.06–22.63]	15.69 [13.67–20.5]	22.82 [16.02–29.11]	22.97 [15.9–28.87]	KW	<0.001
ODose30	16.74 [13.05–25.45]	18.37 [15.28–20.04]	18.8 [16.7–26.47]	22.91 [19.66–29.2]	23.81 [18.21–29.06]	KW	<0.001
ODose45	15.73 [12.91–22.59]	18.08 [15.39–20.26]	18.37 [15.86–22.37]	19.92 [17.25–26.02]	21.84 [18.84–27.41]	KW	<0.001
ODose60	13.48 [11.14–20.5]	15.43 [14.69–17.48]	16.12 [13.82–19.19]	15.79 [13.9–17.55]	17.02 [13.17–22.65]	KW	<0.001
ODose120	8.37 [3.17–14.26]	9.34 [6.27–11.24]	9.84 [6.69–10.6]	8.7 [7.09–13.03]	10.03 [8.01–14.86]	KW	0.033
cODose15	2.0 [1.12–3.15]	2.25 [1.1–3.9]	2.07 [1.11–3.75]	2.8 [2.16–5.89]	3.32 [1.29–4.72]	KW	<0.001
cODose30	7.36 [4.39–9.68]	6.87 [5.08–8.09]	7.29 [5.94–8.38]	8.32 [6.22–12.17]	9.0 [6.18–10.62]	KW	<0.001
cODose45	11.32 [9.52–15.85]	9.48 [8.8–14.18]	9.54 [8.36–14.3]	13.64 [8.75–18.25]	13.81 [10.12–18.83]	KW	0.002
cODose60	15.58 [12.39–19.69]	15.72 [14.2–17.91]	17.14 [15.05–20.23]	17.88 [16.2–21.38]	18.49 [15.24–22.84]	KW	<0.001
cODose120	25.7 [20.43–34.07]	28.02 [24.71–32.4]	27.64 [26.22–35.11]	30.6 [26.35–33.73]	31.85 [27.6–39.98]	KW	<0.001

SD—Standard Deviation; BMI—Body Mass Index; HOMA—Homeostatic Model Assessment for Insulin Resistance; ALT—Alanine Aminotransferase; AST—Aspartate Aminotransferase; ALB—Albumin; INR—International Normalized Ratio; γGT—Gamma Glutamil Transpeptidase; CRP—C Reactive Protein, MDose—PDR for Methacetin; cMDose—cPDR for Methacetin; Odose—PDR for Octanoate; cODose—cPDR for Octanoate, ANOVA—One Way ANOVA test, KW—Kruskal Wallis Test, Chi—Chi Square Test.

**Table 2 jcm-12-02158-t002:** Extended patient characteristics, by study group.

Variable	Group 1	Group 2	Group 3	Method	*p*-Value
Number	31 (38%)	11 (14%)	39 (48%)		
Age	47.26 (SD = 15.11)	48.55 (SD = 11.6)	52.67 (SD = 9.77)	ANOVA	0.179
BMI	28.13 (SD = 5.94)	33.18 (SD = 6.08)	30.92 (SD = 5.29)	ANOVA	0.024
Glucose	107.0 [66–168]	110.0 [78–173]	117.0 [69–205]	KW	0.065
HOMA	2.85 [1.7–3.69]	2.85 [2.56–3.89]	3.18 [2.12–5.19]	KW	0.03
Triglycerides	174.0 [130–293]	163.0 [99–315]	201.0 [74–347]	KW	0.013
Cholesterol	212.0 (SD = 94.0)	234.0 (SD = 63.0)	223.0 (SD = 57.0)	ANOVA	0.662
HDL	35.55 (SD = 6.6)	38.18 (SD = 7.47)	34.0 (SD = 8.82)	ANOVA	0.285
ALT	57.0 [21–133]	78.0 [54–204]	98.0 [46–158]	KW	<0.001
AST	74.0 [48–117]	83.0 [51–127]	84.0 [51–138]	KW	0.063
ALB	4.0 [2–5]	3.0 [3–4]	3.0 [2–4]	KW	0.001
INR	1.0 [0.9–1.3]	1.0 [0.8–1.1]	1.0 [0.9–1.3]	KW	0.455
γGT	77.0 [38–121]	122.0 [41–185]	114.0 [69–279]	KW	<0.001
FERITIN	146.0 [111–203]	154.0 [119–228]	162.0 [106–390]	KW	0.039
CRP	2.06 [0.6–5.71]	5.52 [0.63–7.03]	5.33 [1.99–8.46]	KW	<0.001
URIC ACID	4.88 [3.06–7.26]	6.09 [4.51–9.85]	6.84 [2.84–9.35]	KW	<0.001
Spleen Diameter	104.0 (SD = 11.0)	110.0 (SD = 11.0)	121.0 (SD = 13.0)	ANOVA	<0.001
Abdominal circumference	89.0 [65–104]	95.0 [88–107]	93.0 [64–131]	KW	0.01
Sex	22 F/9 M	6 F/5 M	17 F/22 M	Chi	0.072
MDose10	32.87 [10.67–43.06]	32.62 [27.68–37.84]	16.34 [5.36–43.77]	KW	<0.001
MDose20	29.21 [8.73–36.31]	30.74 [26.34–36.45]	11.36 [8.35–52.75]	KW	0.001
MDose30	23.72 [14.16–30.57]	25.47 [20.63–27.61]	15.12 [11.81–35.1]	KW	0.29
MDose40	19.55 [16.08–27.91]	19.54 [16.61–26.33]	19.08 [13.66–32.33]	KW	0.777
MDose50	15.77 [13.84–23.25]	17.2 [13.71–22.72]	14.04 [11.34–25.94]	KW	0.192
MDose60	13.57 [10.23–17.36]	13.74 [12.33–20.32]	11.74 [8.58–20.15]	KW	0.005
cMDose10	5.3 [2.33–6.11]	5.05 [4.61–6.2]	3.11 [2.51–10.18]	KW	<0.001
cMDose20	8.5 [3.36–10.82]	7.76 [6.34–9.49]	4.27 [3.12–13.7]	KW	<0.001
cMDose30	13.12 [3.96–15.15]	13.29 [11.95–14.6]	6.42 [3.34–21.61]	KW	<0.001
cMDose40	15.68 [4.82–17.96]	15.14 [14.26–16.36]	9.26 [5.33–25.61]	KW	<0.001
cMDose50	18.47 [7.33–21.17]	17.74 [17.04–19.19]	12.15 [8.76–26.96]	KW	<0.001
cMDose60	21.7 [7.94–24.83]	20.68 [17.7–23.94]	13.69 [9.27–30.66]	KW	<0.001
ODose15	19.83 [8.39–22.63]	20.68 [12.59–24.68]	21.42 [13.67–29.11]	KW	0.014
ODose30	16.86 [13.05–25.45]	17.66 [15.78–25.35]	22.57 [16.7–29.2]	KW	<0.001
ODose45	17.95 [12.91–22.44]	17.52 [13.25–22.59]	21.16 [15.86–27.41]	KW	<0.001
ODose60	14.36 [11.14–20.5]	14.69 [11.28–17.36]	16.24 [13.17–22.65]	KW	<0.001
ODose120	9.18 [3.17–14.26]	8.42 [7.02–11.52]	9.75 [6.69–14.86]	KW	0.056
cODose15	2.29 [1.1–3.9]	1.83 [1.22–3.15]	2.85 [1.11–5.89]	KW	0.005
cODose30	7.17 [4.39–9.68]	7.45 [5.77–9.1]	8.47 [5.94–12.17]	KW	<0.001
cODose45	11.31 [8.92–15.85]	10.94 [8.8–14.43]	12.64 [8.36–18.83]	KW	0.036
cODose60	15.63 [12.68–19.69]	16.09 [12.39–18.59]	18.14 [15.05–22.84]	KW	<0.001
cODose120	26.08 [20.43–34.07]	27.91 [20.91–30.96]	30.97 [26.22–39.98]	KW	<0.001

SD—Standard Deviation; BMI—Body Mass Index; HOMA—Homeostatic Model Assessment for Insulin Resistance; ALT—Alanine Aminotransferase; AST—Aspartate Aminotransferase; ALB—Albumin; INR—International Normalized Ratio; γGT—Gamma Glutamil Transpeptidase; CRP—C Reactive Protein, MDose—PDR for Methacetin; cMDose—cPDR for Methacetin; Odose—PDR for Octanoate; cODose—cPDR for Octanoate, ANOVA—One Way ANOVA test, KW—Kruskal Wallis Test, Chi—Chi Square Test.

**Table 3 jcm-12-02158-t003:** Fibrosis and activity crosstab.

Variable	F0	F1	F2	F3	F4	Total
A0	20	6	0	0	0	26
A1	5	0	0	0	0	5
A2	5	2	5	10	12	34
A3	2	2	3	1	7	15
A4	0	0	0	1	0	1
Total	32	10	8	12	19	81

Chi Square = 60.018, *p*-value < 0.001.

**Table 4 jcm-12-02158-t004:** AUROC for different parameters.

	NASH	Significant Fibrosis	Cirrhosis
MDose10	0.800	0.860	0.869
MDose20	0.662	0.726	0.866
MDose30	0.548	0.596	0.793
MDose40	0.495	0.520	0.745
MDose50	0.552	0.588	0.742
MDose60	0.619	0.702	0.811
cMDose10	0.791	0.833	0.844
cMDose20	0.817	0.848	0.823
cMDose30	0.825	0.891	0.889
cMDose40	0.823	0.868	0.895
cMDose50	0.808	0.849	0.897
cMDose60	0.818	0.825	0.858
ODose15	0.652	0.687	0.786
ODose30	0.855	0.887	0.882
ODose45	0.760	0.774	0.879
ODose60	0.754	0.776	0.880
ODose120	0.611	0.652	0.778
cODose15	0.648	0.711	0.791
cODose30	0.734	0.783	0.889
cODose45	0.624	0.666	0.754
cODose60	0.789	0.763	0.882
cODose120	0.756	0.794	0.807

MDose—PDR for Methacetin; cMDose—cPDR for Methacetin; Odose—PDR for Octanoate; cODose—cPDR for Octanoate.

**Table 5 jcm-12-02158-t005:** Delong test for the best parameters.

	Test 1	Test 2	z	*p*-Value
NASH	ODose30	CMDose30	9.091	<0.001
Significant Fibrosis	CODose30	CMDose30	9.470	<0.001

cMDose—cPDR for Methacetin; Odose—PDR for Octanoate; cODose—cPDR for Octanoate

## Data Availability

The data are not publicly available due to the contract signed between institutions participating in the experiment and patients’ confidentiality agreements. The data presented in this study are available upon request from the corresponding author.

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
