# Peer review of "A Comparison of 13C-Methacetin and 13C-Octanoate Breath Test for the Evaluation of Nonalcoholic Steatohepatitis"

_jcm, 2023, doi:10.3390/jcm12062158_

Round 1

Reviewer 1 Report

The authors present a straightforward and interesting study of the utility of breath tests in correlation with liver biopsy findings among patients with NAFLD.  The topic is of broad interest and the work here has significant merit, but there are some outstanding issues that require attention, as below: 

Remove section 0

Line 28-29, section1:

                Description of simple steatosis as a non-progressive entity is an over-simplification and doesn’t reflect the level of uncertainty in its prognosis (https://www.ncbi.nlm.nih.gov/pmc/articles/PMC7698018/). This text should be clarified for the reader. 

Section 1 lines 34-35  Please provide the reader more context regarding estimated mortality risk from liver biopsy. This important tool should be described with more precision to avoid inaccurate perception of risk. I would also recommend better qualifying the statement “rarely accepted by the patient.”

There appear to be numerous instances of unnecessary hyphenation, perhaps from prior line-breaks?

Section 1: Extra period at end of Aim

Abd US description of steatosis qualification appears to be redundant (lines 111-116)

Section 2: Avoid use of term gender, favor “sex”

Regarding definition of Group 3 NASH with "advanced" fibrosis.  Most readers would object to the use of advanced fibrosis for F2, rather than F3+. This terminology is inconsistent with generally accepted definitions and should be altered.  

Can the authors provide more specific data on the interval from liver biopsy to primary testing dates?  

Section 3:

Please specify in supplementary table the reasons for patient exclusion.

Of the 159 patients initially screened, were these consecutive patients seen in a practice who had NASH defined via biopsy? If not, how was this cohort defined?

Section 4:

The authors claim an advantage of this method is real-time available of results “allowing fast-clinical intervention when necessary.” One presumes this is a purported advantage in comparison to liver biopsy (which may take perhaps days or weeks to review depending upon local expertise). It is challenging to imagine a scenario in which a time difference of that scale would be clinically relevant when options for intervention are essentially limited to lifestyle change long-term medical therapy, ambulatory surgical referral, etc.  Can the authors please elaborate on this?

Sections 4/5:
The authors comment on the potential for use of 13C MBT for prediction of increased portal resistance, however they have present no data to directly support this hypothesis and the claim relies too heavily on extrapolation of multiple surrogate markers. While a very interesting potential use case for this agent, I think this statement needs to be tempered to reflect the limits of surrogate markers here.

Notes: 

The authors include no description of the potential limitations of the study, which is of high interest to readers. This must be addressed. For example (among others), exclusion of patients with A1c > 7% significantly limits applicability to all patients with NASH in clinical practice. This should be noted. 

A key concern is whether the inflammatory activity or the fibrosis changes the measurements here.  The biochemical test results here vary with fibrosis and NASH, however fibrosis may vary with NASH activity, as well, suggesting possibility for confounding. A subgroup analysis in which all patients with a given fibrosis stage are assessed for difference in results by inflammatory activity score and/or all patients with a given inflammatory activity score and assessed across stages of fibrosis might help to elucidate this and provide the reader valuable information about what these tests are truly assessing.

The authors repeatedly describe the cohort as patients with NASH and yet the biopsy results show that roughly one-third of patient did not appear to meet this criterion. Please clarify. 

Author Response

Dear Reviewer, please see attachment.

Thank you.

Reviewer 2 Report

Title A Comparison of 13C-Methacetin and 13C-Octanoate 2 Breath Test for the Evaluation of Nonalcoholic Steatohepatitis. My comments are as follows:

1.      Abstract is missing in the article

2.      Please correct words eg. pa-tient, maxi-mum etc

3.      Do the authors have data on Fibroscan values?

4.      Tables can have better representation- remove the SD from every row and can be written in the first column. Method also can be removed as a column and written in footnotes.

5.      Provide analysis data of parameters based on groups 1,2 and 3 as mentioned in the section study groups

6.      Under the diagnostic performance of the 13C breath tests provide the ROC curves for diagnosis of NASH. Consider providing data for ROC predicting significant, advanced fibrosis and cirrhosis.

7.      Also compare the 2 tests- AUROCs with DeLong tests to show if there is any difference in the AUROCs of the 2 tests.

8.      The authors have written the term "conclusion" twice. Better to write it once

9.      MAFLD a term out of context in this paper. Consider removing it

10.  Write limitations section of the paper.

11.  Small sample size is a major limitation.

12.  Gap of 1-year between the liver biopsy and breath tests is a major limitation of the study, changes in NASH would have happened over this period.

13.  Provide percentages for categorical variables in the tables.

Author Response

Dear Reviewer, 

Thank you.

Round 2

Reviewer 2 Report

None

Author Response

Thank you very much.